# Alleviating the Bauxite Maritime Supply Chain Risks through Resilient Strategies: QFD-MCDM with Intuitionistic Fuzzy Decision Approach

**Jiachen Sun** [1,†]**, Haiyan Wang** [1,2,†]  **and Zhimin Cui** [1,*]

1    School of Transportation and Logistics Engineering, Wuhan University of Technology, Wuhan 430063, China; sunjiachen1969@163.com (J.S.); hywang777@whut.edu.cn (H.W.)
2    National Engineering Research Center for Water Transport Safety, Wuhan University of Technology, Wuhan 430063, China
*    Correspondence: zmcui1213@whut.edu.cn
†    These authors contributed equally to this work.

**Abstract:** With the development of the global economy and energy supply chain, the uncertainty and complexity of the bauxite maritime supply chain (BMSC) has been further increased. Determining the crucial risks and improving the supply chain's resilient capacity based on operation objectives has become important, in order to ensure the sustainability and competitiveness of the BMSC. This paper combines quality function deployment (QFD), a multi-criteria decision method (MCDM), and intuitionistic fuzzy set (IFS); an integrated methodology is developed to achieve efficient design of BMSC resilient strategies (RESs), taking into account both customer requirements (CRs) and risk factors (RFs). A combined weighting method is employed to determine each CR's importance. A decision-making trial and evaluation laboratory (DEMATEL) method is adopted to determine the RFs' interrelationships. The results obtained with the MCDM are incorporated into QFD to construct a two-stage house of quality (HoQ) model, which transforms CRs into RFs, and then into RESs. The real case of the Guinea–China bauxite import supply chain is studied to demonstrate the applicability and validity of the proposed framework. Research results reveal that the most important CR is 'stability'. 'Information sharing asymmetry', 'poor ship stability or obsolete equipment performance', and 'lack of coordination between shipping and ports' are the most severe risks impacting the operation of supply chain. Furthermore, 'constructing strategic alliances' contributes to alleviating potential risks, optimizing the allocation of resources, and finally, improving the resilience of the BMSC significantly. This paper will help managers to understand how to achieve sustainable development of the supply chain through resilient strategies, and will aid rational decision-making in the management and operation of a resilient BMSC for alleviating risk.

**Keywords:** bauxite maritime supply chain (BMSC); resilience; risk; QFD; DEMATEL; intuitionistic fuzzy set

## 1. Introduction

Bauxite is an important energy basis for national economic construction, and it plays an important role in ensuring the safety and sustainability of the industrial chain and supply chain. It underpins the rapid development of the aluminum industry, refractory materials, corundum abrasives and high alumina cement industries. With the challenge of satisfying the world's growing energy demand, the consumption of bauxite is predicted to show a steady upward trend in various countries. According to the data of World Mineral Summary 2021, released by the US Geological Survey, global bauxite resources are distributed throughout Africa (32%), Oceania (23%), South America (21%), Asia (18%) and other regions (6%) [1]. Because of the uneven distribution of bauxite mines around the world, most countries' external dependence on bauxite is increasing year by year. China,

for example, ranks seventh, with 1 billion tons of bauxite reserves, accounting for 3% of the world's bauxite reserves. In 2020, the import volume of bauxite hit a record high of 110 million tons, and the external dependence increased from 5% in 2001 to 50% [2]. Therefore, shipping and specialized ports are needed to transport and process bauxite over long distances. Maritime transportation is the major method of bauxite transportation due to its great economy and ability to move large volumes; in this context, the bauxite maritime supply chain (BMSC) is formed. A typical BMSC is illustrated in Figure 1. It starts with the extraction of the bauxite mine onshore, which is processed and stored before being concentrated transported to loading port by different participants using various modes of transportation. Over longer distances, subsequent transport overseas uses specialized vessels to move the bauxite to the receiving terminal. At the destination terminal, the bauxite is temporarily processed at the port, before being dispersed to chemical enterprises and other customers.

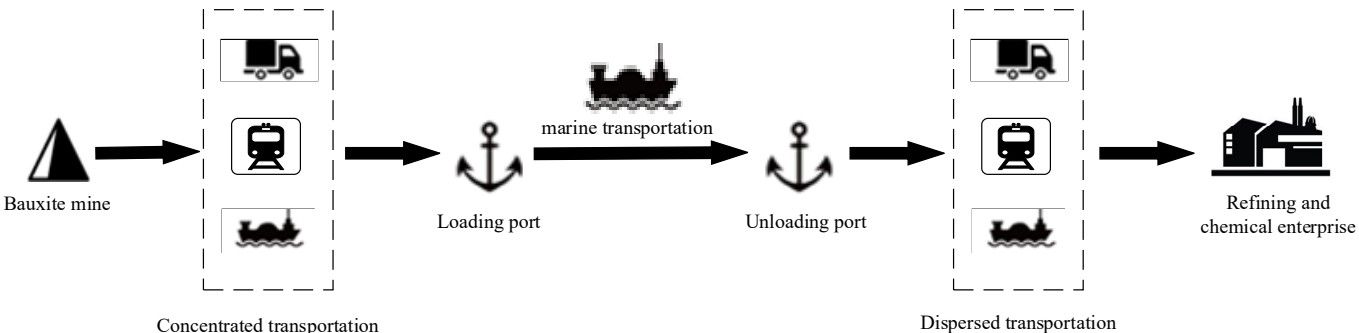

**Figure 1.** A typical bauxite maritime supply chain.

In the BMSC, the supply and demand are connected by a logistics chain, in which logistics involves inland transportation, terminal operations and maritime transportation. Supply chain participants conduct business through cooperation and information exchange. With the influence of uncertain disruption events, it has also become extremely difficult for the globalized and complex BMSC to achieve energy efficiency. In recent years, COVID-19, the Russia–Ukraine conflict and other black swan events have been severely and globally disruptive, causing countries to enact border closures, bring production to a standstill, and stop import–export activities. In addition, in the African region, the world's largest exporter of bauxite, supply activity has been severely hampered by unstable political conditions, worker unrest and port strikes. For example, in September 2021, a coup in Guinea led to the suspension of the constitution and the closure of the country's borders, forcing the suspension of the country's bauxite exports. On the flipside, the occurrence of uncertain events leads to increased risks for logistics network, which also highlights the vulnerability of the BMSC. In the BMSC, risks related to logistics involve five aspects: humans, carriers, the environment, goods and management. These may be due to the lack of safety awareness of logistics operators or a lack of emergency response skills needed to face operational errors. Particularly in long-distance maritime transportation, human negligence, the ship's hull structure and integrity should not be ignored. In addition, the fluidity of bauxite is one of the most important reasons for the logistical risk. In 2015, a ship carrying 46,400 tons of bauxite from Kuantan port in Malaysia capsized on route to China because of bauxite fluidization. Due to the poor management of supply chain actors, information sharing asymmetry, and improper resource allocation, other problems often occur. As an important node of BMSC, shipping at ports is often not smooth, resulting in the untimely arrival of goods and cost increases. While in the past, maritime supply chain risk management has focused on cost savings and safety guarantees, today, it is clear that ensuring the resilience of supply chain should generally be the top priority [3]. From the perspective of the supply chain, risks may affect the sustainability and stability of the

bauxite supply, and resilience ability can help to maintain normal performance levels and achieve sustainable development of the BMSC in an uncertain environment.

As more uncertainties are exposed in the decision-making process of supply chain management, in recent years, some experts have proposed that one of the most efficient and powerful methods to deal with supply chain risks is the infusion of resilient strategies [4,5]. In light of this, aiming to satisfy customer requirements and alleviate uncertainties existing in the operation of the BMSC, this paper proposes alleviating the risks of BMSC through resilient strategies, so as to assess the severe risks, ensure the normal flow of bauxite in the supply chain network, and achieve effective management of various activities along the supply chain. As a result, the research questions of this study are summarized as follows:

- RQ1: How can the risks of BMSC be assessed to cope with customer requirements for supply chain operations?
- RQ2: What strategies should be prioritized to improve the resilience of the BMSC, maximize risk resistance and ultimately satisfy customer requirements of supply chain operations in an uncertain environment?

To solve the aforementioned questions, a QFD-MCDM with an intuitionistic fuzzy decision approach is developed. Firstly, the resilient strategies (RESs) of the supply chain from the five aspects of supply, transportation, information, organization are summarized through combining the characteristics of the maritime supply chain and referring to a relevant literature review. Secondly, the customer requirements (CRs) for the BMSC's operations, the potential risk factors (RFs), and the appropriate RESs are identified. In addition, the weight of CRs, the interdependence of RFs, the relationship between CRs and RFs, and the relationship between RFs and RESs are investigated comprehensively. Then, the severity of the RFs and the priority of the RESs in the BMSC are determined. Finally, the BMSC of Guinea–China is used as a case study; the proposed methodology is used to assess RFs and provide managers with useful RESs for alleviating risk impacts.

The contribution of this study involves three aspects: (1) So far, a number of studies on container maritime supply chains have been carried out. However, studies related to the maritime supply chain of bulk cargo, especially bauxite, are insufficient. Therefore, this study enriches the theoretical research into BMSCs. (2) Although some studies have assessed the potential risks of the maritime supply chain from qualitative or quantitative aspects, they have not been combined with the customer requirements of supply chain operations. Furthermore, some scholars have discussed risk mitigation strategies conceptually. However, the research on mitigating risks from the perspective of a designing supply chain with resilience is insufficient. Therefore, this study constructs a two-stage house of quality (HoQ) model based on quality function deployment (QFD) to connect CRs, RFs and RESs. (3) In this study, a multi-criteria decision method and intuitive fuzzy set (IFS) are integrated into QFD, which provides a methodological framework for supply chain risk management and resilient design.

The rest of the paper is organized as follows. Section 2 reviews the relevant literature, the existing research into maritime supply chain risk, and supply chain resilience; the applications of QFD are also summarized Section 3 proposes hybrid methodology integrating QFD, MCDM and fuzzy set theory. Section 4 provides a case study and discussion. Finally, the conclusion and future research are presented in Section 5.

## 2. Literature Review

In this section, maritime supply chain risks, supply chain resilience, the QFD method and its application in supply chain management have been presented. In the following subsection, each of these parts is analyzed separately.

### 2.1. Maritime Supply Chain Risks

Maritime-related industries are vital to international economic and social wellbeing. For every country's trade and commerce, seaports act as critical nodes, and play an important role in facilitating maritime connectivity across the globe [6]. With the rapid de-

velopment of global trade, maritime supply chains have become one of the largest complex networks in the world. Different from ordinary supply chains, maritime supply chains have seaports at their center, and effectively integrate shippers, freight forwarders, inland transport service providers, shipping companies and other service nodes to complete the supply activity of goods [7].

However, the increasingly high degree of coupling and interaction between stakeholders and processes, as well as over-dependence on deeper and broader global chains, has made the operation of maritime supply chains more vulnerable to uncertain disruptive events. According to the findings of the literature review, maritime supply chain risks can be typically classified into two categories, namely disruption risks and operation risks [8,9]. Disruption risk refers to uncertainty and disturbances occurring within the normal process of the supply chain. They might occur as a result of human-related events (pandemics, strikes, low wages/salaries, poor working environments, etc.), technical failures (insufficient facilities in the warehouses, obsolete facilities, slow pace of digitization, etc.), or the external environment (natural disasters, political turmoil, conflicts of law, etc.). On the other hand, operation risks refer to the uncertain interference caused by the cooperation of participants in normal business, for instance, poor communication of information, and lack of coordination and organizational structure.

Wan et al. [9] identified the main risk factors of container maritime supply chain from five perspectives: society, the natural environment, operations, infrastructure and technology, and management. Their study provided useful insights to participants from different parts of the maritime supply chain to better understand the risks in their daily operations from a whole supply chain perspective. Kashav et al. [8] classified risk barriers related to the container maritime supply chain into six categories, namely economic, infrastructural, technological, administrative and political, legal, and organizational. In this paper, the findings indicate that infrastructure and legal barriers are the two most critical categories. Narasimha et al. [7] studied the impact of COVID-19 on port transport and maritime supply chain performance indicators in India, as well as issues in preparation, response and recovery. Their findings contributed to the development of maritime strategies that address risks while enhancing supply chain resilience and sustainable business recovery processes. Zavitsas et al. [10] simulated the impact of waste control on maritime operational safety and correlated it with network resilience performance indicators to minimize disruption risks while minimizing operational costs in the maritime supply chain. Panahi et al. [11] developed an effective risk assessment model to quantify the maritime supply chain risks associated with extreme weather events in the Arctic. The study provided useful enlightenment for all parties and stakeholders in maritime transport. Yang et al. [12] established a game model based on Bayesian networks, considered the impact of inspection risks on port state control, and determined the optimal inspection strategy of port authorities. As an important part of supply chain activities, maritime logistics are a form of value-added supply chain management that integrate ocean transportation and integration of inland or inland ports. Jia et al. [13] summarized the information risks of maritime logistics from the three aspects of information accuracy, timeliness and security, and the operation risks of maritime logistics from the six aspects of transportation delay, cargo damage, asset loss, customs clearance, storage and personnel safety. In the study, the soft set theory was used to reveal the impact of inaccurate information transfers on transportation delay risks and storage risks. In the context of the maritime silk road, Jiang et al. [6] adopted fuzzy logic and evidential reasoning methods to assess the risks of the maritime supply chain, introduced multi-dimensional parameters from different perspectives, and prioritized the critical security risks in the uncertain environment.

It is clear from the above analysis that existing research efforts have recognized the significance of the risks in maritime supply chains. However, most of the existing literature related to maritime risks focuses on analyzing and assessing the risks of ship transport and operation; moreover, it being mainly related to containers, research related to bulk cargo

has been lacking. In addition, studying from the perspective of the whole supply chain is an important research direction.

*2.2. Resilient Supply Chains*

With the globalization of supply chain, the distance between suppliers and customers has increased, and with the influence of interference events, the supply chain has become more complex. More uncertainties have been exposed in the decision-making processes of supply chain management. Therefore, global supply chains are more vulnerable to risks due to their complex structures. One of the most efficient and powerful methods to deal with disruption risks is designing resilient networks [10,14].

In the field of engineering, Youn et al. [15] regarded resilience as the sum of the passive survival (reliability) and active survival (resilience) of a system, which refers to the inherent ability of a system to adjust its function in the face of interference or unanticipated changes. The study of supply chain resilience began with the British public's protests in 2000 over rising oil prices, and the supply chain disruptions caused by the outbreak of foot and mouth disease in Britain in 2001. The global supply chain is faced with an increasingly complex, dynamic and uncertain operating environment, and the resilient supply chain, which mainly aims to cope with supply chain disruption, has attracted more and more attention from scholars. A few studies have limited supply chain resilience to disruption caused by natural disasters, believing that resilience refers to the ability to quickly restore operations and maintain normal functions and structures after disasters. More studies believe that supply chain disruption is general and universal, not only referring to catastrophic events such as earthquakes, floods, terrorist attacks, etc., but also to operational changes and risks, such as supply shortages, production line stagnation, customer order cancellations, etc. Recently, extensive studies have been conducted on supply chain disruptions and the concept of resilience. One of the most efficient and powerful methods to deal with risks is integrating resilient strategies; further, a resilient supply chain would be more resistant to potential risks [16].

A large number of studies have described the important attributes of a resilient supply chain. According to the process of supply chain interruption, a resilient supply chain can be divided into the preparation stage before distribution, the response stage during distribution, and the recovery stage after distribution. According to its ability to cope with supply chain disruption, it is considered that resilient supply chain should have the characteristics of robustness, redundancy, agility, adaptability, visibility and so on. For instance, Ali et al. [17] pointed out that a resilient supply chain should have the following five capabilities: prevention, resistance (i.e., increasing redundancy and flexibility), response, recovery, learning and continuous improvement.

Azadeh et al. [18] evaluated system resilience from four aspects, visibility, redundancy, flexibility and recovery speed, in a three-level supply chain, considering transport disruption, and used fuzzy data envelopment analysis to determine the optimal scheme, including the elasticity index. The results showed that visibility and redundancy played an important role in transport elasticity factors. Piya et al. [19] identified factors affecting the resilience of oil and gas supply chains during the pandemic; the fuzzy ISM-DEMATEL method was used to analyze these driving factors. The results showed that government regulations and safety are the most important driving factors, followed by information sharing, effective cooperation, and risk management culture. Flexibility and robustness are the key factors which are affected by other factors and then affect the overall performance of resilience. Namdar et al. [20] explored procurement strategies in improving supply chain resilience, including single and multiple purchasers, procurement visibility, collaborative procurement, etc., and developed a scenario-based mathematical model to assess the impact of different strategies on supply chain resilience. The results showed that procurement of early warning capabilities is critical for enhancing supply chain resilience. Belhadi et al. [21] studied the resilience of manufacturing and service supply chains during the pandemic, and studied strategies to improve supply chain resilience. The results showed

that strengthening information sharing, supply chain collaboration and the supply chain's digital technology is of great significance for improving the supply chain resilience of the automobile and aviation industries. Shao et al. [22] adopted a dynamic analysis method of system dynamics modeling to evaluate the resilience level of the lithium supply chain, by analyzing the multiple responses to different interruptions caused by new energy vehicles. They analyzed resilience mechanisms from the three perspectives of supply, price, and demand, and the simulation results showed that the greater the demand shock, the longer the supply disruption, the lower the resilience of the lithium supply chain, and the lower the impact of material substitution on the resilience.

Considering that resilience has important research and practical value for supply chain risks, this paper summarizes the diversified resilient strategies proposed by scholars to cope with different situations, as shown in Table 1.

**Table 1.** Summarization of the RESs of the supply chain.

| Resilient Aspects | Resilient Strategies (RESs) | Source |
|---|---|---|
| Supply level | Dual and multiple sources of supply | [20] |
| | Keeping back-up suppliers | [20,23] |
| | Appropriate inventory management | [23] |
| | Providing resource investment to suppliers | [3] |
| | Strategic sourcing, flexible sourcing and order fulfillment | [24] |
| | Selecting suppliers close to the place of production | [5,16] |
| Transportation level | Multimodal transportation, multicarrier transportation, and multiple routes | [18] |
| | Internal and external logistics collaboration | [5] |
| | Logistics operation flexibility, traceability and digitization | [25] |
| | The optimization of logistics infrastructure and personnel capacity | [26] |
| | Flexibility to change delivery routes | [27] |
| | Developing robust, reliable logistics facilities | [5,27] |
| Information level | Seamless information sharing | [28,29] |
| | Implementing an integrated database in internal supply chains | [28] |
| | Early warning communication | [30] |
| | Making the supply chains visible and transparent using blockchain technology, big data and digital technologies | [25] |
| Organization level | Collaboration and co-opetition among supply chain stakeholders | [31–33] |
| | Vertical and horizontal integration of the supply chain | [3,34] |
| | Security, monitoring, early warning and maintenance | [30] |
| | Supply chain relationship management | [32,35] |
| | Skill and efficiency development through training and counselling | [30] |
| | Knowledge and risk management culture | [24] |

### 2.3. QFD and Its Application in Supply Chain Management

QFD is a systematic method used to transform customer requirements (CRs) into design requirements (DRs) of products, processes, services, and strategies that achieves the goal of improving process quality, reducing cost and enhancing customer satisfaction. Because of its wide applicability, QFD has been used in various fields [36–38], such as determining customer needs, developing priorities, manufacturing strategies, logistics and supply chain management. HoQ is the key technology of the QFD method; it can effectively show CRs and their importance, DRs and their autocorrelation, and the correlation between CRs and DRs. The HoQ consists of six main parts: (1) CRs; (2) DRs; (3) CRs' importance; (4) the relationship matrix among DRs; (5) the relationship matrix between CRs and DRs; and (6) the satisfaction output of DRs. The structure of HoQ is shown in Figure 2.

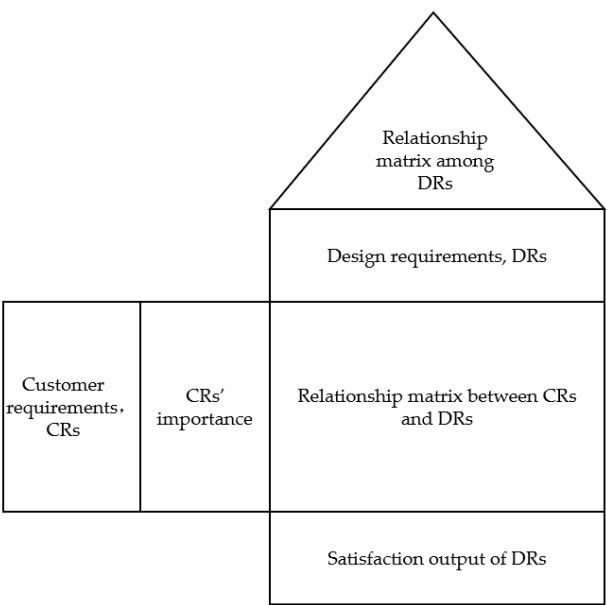

**Figure 2.** The structure of HoQ.

In the literature, the QFD method and its improved methods have been applied to the research into supply chain management. For instance, Chowdhury et al. [39] developed a 0–1 multi-objective optimization model based on a QFD methodology to mitigate ready-made garment supply chain vulnerabilities. In the study, the vulnerabilities of the supply chain and the most satisfactory efficient portfolio resilience strategies were established. Mohamed Abdel-Basset proposed a combination of QFD with lithogenic aggregation operations for selecting sugar industry supply chain sustainability metrics. The study revealed that the introduced approach can be employed to alleviate the degree of difficulty of information gathering, and to choose the most preferred metric. Original QFD has been considered a manual approach and has also been known for having several limitations, such as a long implementation time and subjective decision aid. In some studies, the MCDM method is combined with QFD to solve interactive decision problems. Ramezankhani et al. [40] developed a hybrid method using QFD together with decision-making trial and an evaluation laboratory (DEMATEL) to apply to the automotive manufacturing sector. In the paper, the best sustainability and resilience factors were selected systematically. Hsu et al. [41] developed an integrated framework of QFD and MCDM to devise an effective method to mitigate sustainable supply chain risks by improving supply chain agility. The results showed that the proposed framework can be effectively used by electronics manufacturers to develop agile strategies to mitigate sustainable supply chain risks. He [42] utilized a Kano model and DEMATEL integrated with QFD through nonlinear programming to guide sustainable supply chain design. In the study, DEMATEL was adopted to determine the risk factors' interrelationships, the Kano model was employed to determine CRs, and the QFD method was used to identify relationships between CRs and resilience measures. Besides these integrations, QFD has generally been used with the fuzzy set theory to consider vagueness in linguistic evaluation. An integrated MCDM-based QFD with hesitant fuzzy linguistic term sets was adopted by Erol et al. [43] to explore blockchain technology's ability to solve the problems of circular economy at the supply chain level, and a ranking of strategies was determined. Haiyun et al. [44] analyzed innovation strategies for green supply chain management with QFD; an interval-valued intuitionistic fuzzy set was used to collect and evaluate some data on the subject, under uncertain conditions.

## 3. Methodology

A QFD approach integrated with an analytic hierarchy process (AHP), an entropy weight method (EWM), DEMATEL and IFS is proposed in this paper to incorporate CRs

into supply chain risk resilience management and achieve an effective BMSC resilience solution to minimize critical risks. This methodology will help managers to analyze serious RFs and their interrelationships, and determine the optimal RESs and their associations, both of which may be used to proactively build a resilient network, reduce risks and satisfy CRs. Here, we graphically present the overview of the entire methodology before describing it in detail, as shown in Figure 3. It is divided into three stages, namely customer requirements analysis, risk factors analysis and resilient strategies analysis. The proposed approach begins with determination of the weight for each CR. Secondly, the first HoQ is constructed to identify the relationships between CRs and RFs, and determine the initial severity of RFs. Then, the DEMATEL is applied to explore the casual relationships between RFs to determine the influence importance of RFs, which is integrated with the first HoQ to obtain the final severity of RFs. Thirdly, the second HoQ is established to model the relationships between RFs and RESs, and determine each RES's priority. In order to address the inherent uncertainty of the relationship between things, the intuitionistic fuzzy set is used to reflect the fuzziness of the decision-making environment.

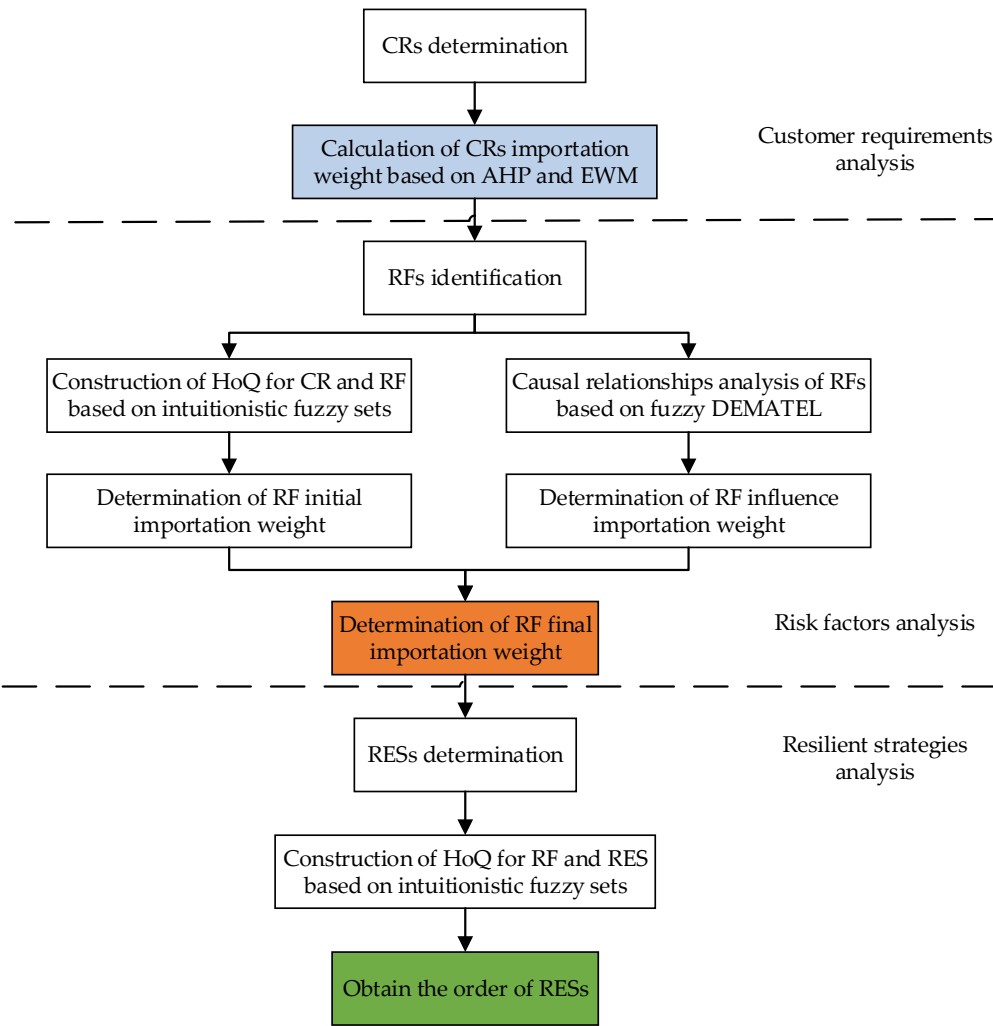

**Figure 3.** Proposed QFD-MCDM with intuitionistic fuzzy linguistic term set framework.

### 3.1. Determination of CRs, RFs and RESs

It is crucial to improve the competitiveness of the supply chain through identifying customer requirements correctly and providing appropriate services to achieve the operational objectives of the supply chain. This study systematically reviews the literature and case reports related to maritime supply chain, starting from identifying the CRs of the BMSC. Taking into account the characteristics of both the supply chain [45] and maritime

industries [37] combined with the BMSC operation processes and business situation, and fully consulting experts' opinions from related fields, the CRs are summarized as reliability CR1, punctuality CR2, security CR3, stability CR4, and economy CR5.

Then, on the basis of the literature review in Section 2.1, according to the risk inspection reports and experts' screening, the RFs of the BMSC are determined from the aspects of humans, infrastructure and technology, the external environment and management. Table 2 shows the detailed list of RFs.

**Table 2.** Risk factors (RFs) of BMSC.

| Category | Risk Factors (RFs) | Notations |
| --- | --- | --- |
| Human risks | Lack of qualified managerial personnel or labor unavailability | RF1 |
| | Operating violations | RF2 |
| Infrastructural and technological risks | Lack of bauxite mine modernization in terms of processed technology | RF3 |
| | Unavailability of dedicated transshipment and feeder port infrastructure | RF4 |
| | Inadequate capacity of operating infrastructure at ports | RF5 |
| | Poor ship stability or obsolete equipment performance | RF6 |
| Environment risks | Unstable government policies and economy | RF7 |
| | International aluminum price and freight market fluctuations | RF8 |
| | Natural disasters (especially terrible sea conditions) | RF9 |
| | Congestion in coastal, waterway or sea areas | RF10 |
| | Sudden security crisis (such as terrorism, pandemic et al.) | RF11 |
| Management risks | No organizational integration in the transportation resources | RF12 |
| | Lack of coordination between shipping and ports | RF13 |
| | Information sharing asymmetry | RF14 |
| | Poor emergency response practices | RF15 |

Furthermore, combined with the above literature summary and analysis on resilient supply chains, most believe that resilient solutions must focus on supply, transportation, information and organization. By reviewing the actual requirements of the BMSC and the critical risks it currently faces, seven RESs are summarized from different aspects. The details are elaborated upon in Table 3.

**Table 3.** Resilient strategies (RESs) of the BMSC.

| Notations | Resilient Strategies (RESs) | Description |
| --- | --- | --- |
| RES1 | Maintain multiple supply sources | Select bauxite supply sources geographically, and stratify them to avoid the possibility of simultaneous interference of supply sources; promptly coordinate other emergency supply sources in case of failure risks to maintain the stability and continuity of the BMSC. |
| RES2 | Moderate redundancy management | Add valuable backup and reasonably set up resource redundancy before risk disturbance, including setting up bauxite strategic inventory, standby suppliers, standby transfer and unloading ports, standby carriers, etc., so that the BMSC can rely on the timely calling of standby resources to satisfy customer requirements in case of sudden failure risk. |
| RES3 | Strengthen the capacity of infrastructure at each node | In order to cope with risks in the optimization design of supply chain, reasonable investment and reinforcement of production facilities are carried out at each node, including adding advanced mining equipment or building aluminum plants, regularly repairing port loading and unloading equipment, expanding storage yard capacity in port areas, building inland transfer terminals or railways, etc. |

**Table 3.** *Cont.*

| Notations | Resilient Strategies (RESs) | Description |
|---|---|---|
| RES4 | Improve flexibility of operation | Through flexible supply strategies, timely adjustment of a purchasing plan, flexible transportation strategies, personnel assignment, restructuring of management organization structures, flexible cooperation contracts to ensure the smooth supply of bauxite, transportation and information, so as to increase the flexible ability to cope with external uncertainties and risks. |
| RES5 | Collaboration and information sharing | Develop trusting relationships between nodal enterprises, quickly adapt to changes in the external environment, and work cooperatively to cope with the impact of interference risks. Ensure information exchange and sharing between supply chain nodes, improve the visualization level of supply chain risk monitoring and early warning ability, and capture disturbance information in a timely manner. |
| RES6 | Constructing strategic alliances | The refinery and chemical enterprises may sign joint operation agreements with mines, carriers, port groups and other nodal enterprises, and carry out project cooperation operations such as fundraising, mine joint operation, and transport capacity allocation. Additionally, then, they may carry out horizontal or vertical alliance activities, so as to achieve unified management through technology and resource complementarity, shared risks and benefits. |
| RES7 | Create a risk-resilient management culture | Set up a risk management team to enhance the awareness of risk management, build a risk-resilient management culture covering the whole supply chain, and anticipate various potential supply and transportation risks to the greatest extent. |

### 3.2. Calculating the Weight of CRs

Weight analysis of CRs is an important aspect of this study. In this section, AHP and EWM are employed to determine the relative importance of CRs. To elicit the decision-makers' (DMs) intrinsic preference for CRs, a questionnaire is developed in which the DMs must choose an answer to express their judgment of CRs' importance.

AHP is a MCDM method combining quantitative and qualitative analysis. It determines the weight of system factors through pair-to-pair comparisons and consistency judgments among factors. The specific steps are as follows: (1) The judgment matrix is constructed through a pair-to-pair comparison of the relative importance of CRs among DMs, and the 1–9 scale method is used to compare the importance degree of each CR; (2) The maximum eigenvalue $\lambda_{max}$ of the judgment matrix and the weight $W_j$ of CR are calculated; (3) The consistency test is carried out. If the consistency ratio is less than 0.1, the consistency test is accepted, indicating that the consistency of the judgment matrix is reasonable; otherwise, it needs to be reexamined [46].

EWM can solve the problem that the result of AHP is too subjective by carrying out an objective analysis on the entropy value of the statistical data. The corresponding weights are determined by the amount of information reflected in the various degrees of index data [47]. The specific steps are as follows:

(1) Normalize the original indicator data:

$$Y_{ij} = \frac{G_{ij} - (G_{ij})_{\min}}{(G_j)_{\max} - (G_j)_{\min}} \tag{1}$$

where $Y_{ij}$ is the evaluation value of the $i$th evaluation of the $j$th evaluation indicator after normalization; $G_{ij}$ is the evaluation value of the $i$th evaluation object of the $j$th evaluation indicator; and $(G_j)$max and $(G_j)$min are the maximum and minimum values of the evaluation indicators in row $j$, respectively.

(2) Calculate $P_{ij}$:

$$P_{ij} = \frac{Y_{ij}}{\sum_{i=1}^{k} Y_{ij}} \tag{2}$$

where $P_{ij}$ is the proportion of the $i$th evaluator in the $j$th indicator; $k$ is the number of evaluation objects.

(3) Calculate information entropy:

$$e_j = -\frac{1}{\ln(u)}\sum_{i=1}^{k} P_{ij}\ln(P_{ij}) \tag{3}$$

where $e_j$ is the entropy value of the $j$th evaluation indicator; $u$ is the number of indicators.

(4) Calculate the entropy weight of indicators:

$$H_j = \frac{1 - e_j}{u - \sum_{j=1}^{u} e_j} \tag{4}$$

where $H_j$ is the entropy weight of the $j$th indicator.

In order to reasonably determine the weight and improve the accuracy of the evaluation results, this paper combined the subjective weight $W_j$ calculated by AHP and the objective weight $H_j$ calculated by EWM. The comprehensive weight $D_j$ of CRs is calculated by adopting the Formula (5):

$$D_j = \frac{\sqrt{W_j H_j}}{\sum_{j=1}^{u} \sqrt{W_j H_j}} \tag{5}$$

*3.3. Establishment of Correlation Matrix*

Owing to the complexity and uncertainty of the decision-making environment and the fuzziness of human thinking, in this paper, IFS is used to represent the DMs' evaluation preference information on the correlation degree among indicators. Finally, an aggregation operator is utilized to aggregate evaluation information to obtain the comprehensive correlation degree among indicators, which helps to make up for the shortage of information acquisition and the lack of accurate value sample data. IFS is an extension of fuzzy conventional set, which includes the membership functions, non-membership functions and hesitant edge groups. IFS data are more comprehensive than a fuzzy conventional set with membership functions only, and can deal with uncertain information more flexibly [44,48]. In view of this, IFS is used to transform the linguistic description into exact values, and finally, to determine the correlation matrix.

**Definition 1.** *Considering X as a fixed set, intuitionistic fuzzy A in X is introduced:*

$$A = \{\langle x, u_A(x), v_A(x) | x \in X \rangle\}$$

*where, $u_A(x) : X \to [0,1]$ and $v_A(x) : X \to [0,1]$ are denoted as the degree of membership and non-membership functions, respectively, and satisfy $0 \le u_A(x) + v_A(x) \le 1$, $x \in X$. $\pi_{\widetilde{\alpha}} = 1 - \mu_{\widetilde{\alpha}} - v_{\widetilde{\alpha}}$ is denoted as the hesitancy of x belonging to the intuitionistic fuzzy set A. Additionally, if $\widetilde{\alpha} = (\mu_{\widetilde{\alpha}}, v_{\widetilde{\alpha}})$, $\widetilde{\alpha}_1 = (\mu_{\widetilde{\alpha}_1}, v_{\widetilde{\alpha}_1})$, $\widetilde{\alpha}_2 = (\mu_{\widetilde{\alpha}_2}, v_{\widetilde{\alpha}_2})$ are three intuitionistic fuzzy numbers and $\lambda$ is a real number, the following are three corresponding arithmetical operations:*

$$\widetilde{\alpha}_1 \oplus \widetilde{\alpha}_2 = (\mu_{\widetilde{\alpha}_1} + \mu_{\widetilde{\alpha}_2} - \mu_{\widetilde{\alpha}_1}\mu_{\widetilde{\alpha}_2}, v_{\widetilde{\alpha}_1}v_{\widetilde{\alpha}_2})$$

$$\widetilde{\alpha}_1 \otimes \widetilde{\alpha}_2 = (\mu_{\widetilde{\alpha}_1}\mu_{\widetilde{\alpha}_2}, v_{\widetilde{\alpha}_1} + v_{\widetilde{\alpha}_2} - v_{\widetilde{\alpha}_1}v_{\widetilde{\alpha}_2})$$

$$\lambda\widetilde{\alpha} = (1 - (1 - \mu_{\widetilde{\alpha}})^{\lambda}, v_{\widetilde{\alpha}}^{\lambda}), \lambda > 0$$

Set $\widetilde{\alpha}_i = (\mu_{\widetilde{\alpha}_i}, v_{\widetilde{\alpha}_i}), (i = 1, 2, \ldots, n)$ as $n$ intuitionistic fuzzy numbers, and aggregate them using a weighted average (IFWA) operator:

$$IFWA_\omega(\widetilde{\alpha}_1, \widetilde{\alpha}_2, \ldots, \widetilde{\alpha}_n) = \omega_1\widetilde{\alpha}_1 \oplus \omega_2\widetilde{\alpha}_2 \oplus \cdots \oplus \omega_n\widetilde{\alpha}_n \tag{6}$$

where $\omega = (\omega_1, \omega_2, \cdots \omega_n)^T$ is the weight corresponding to n intuitionistic fuzzy numbers.

Using defuzzification of Equation (7), the exact value of the intuitionistic fuzzy number is obtained [49].

$$I(\widetilde{\alpha}) = \frac{\exp\left\{\mu_{\widetilde{\alpha}} - v_{\widetilde{\alpha}} + (\mu_{\widetilde{\alpha}} - v_{\widetilde{\alpha}})^3 \pi_{\widetilde{\alpha}}\right\}}{1 + \pi_{\widetilde{\alpha}}} \tag{7}$$

The correlation matrix is an important part of constructing an HoQ model. It describes the correlation degree among indicators. There are $k$ DMs, and their weight is $\{w_i | i = 1, 2, \ldots k\}$. The intuitionistic fuzzy linguistic term in Table 4 is referenced to evaluate the correlation between CRs and RFs, RFs and RESs. After weighting the defuzzification using Equations (6) and (7), the correlation matrices Cum and Emo are obtained, respectively. Cum is the correlation between CRs and RFs, indicating the degree of impact of RFs on CRs; Emo is the correlation between RFs and RESs, and represents the degree of mitigation of RFs by RESs. $u$, $m$, and $o$ are the number of CRs, RFs, and RESs, respectively.

**Table 4.** Degree of correlation and the corresponding fuzzy number.

| Degree of Correlation | Intuitionistic Fuzzy Values |
|---|---|
| Very Strong (VS) | (0.95, 0.05, 0.00) |
| Strong (S) | (0.80, 0.15, 0.05) |
| Moderate (M) | (0.65, 0.25, 0.10) |
| Weak (W) | (0.50, 0.35, 0.15) |
| Very Weak (VW) | (0.40, 0.40, 0.20) |

### 3.4. Influence-Importance Analysis of RFs

Most RFs of the BMSC are often interrelated in practice, and their interrelationship may affect their severity decision-making. Therefore, it is of great importance to unearth the interrelationships and integrate them in the severity analysis. In this section, DEMATEL is utilized to explore the intensity of interaction between RFs, and IFS are used to adequately express the evaluation information of DMs. DEMATEL is a well-known method for studying complex relationships between interdependent factors. It calculates the impact degree and influence degree of each factor on other factors, based on the logical relationship and direct influence matrix among factors in the system [36,42]. The calculation results provide decision support for supply chain managers to focus on the comprehensive severity of RFs. An influence-importance analysis of RFs based on fuzzy DEMATEL is shown in the following steps.

(1)　Establishment of the direct-relation matrix *B*.

$k$ experts applied the intuitional fuzzy terms in Table 5 to evaluate the interaction between RFs, converted the evaluation values of fuzzy language terms into numerical evaluation values, and obtained the fuzzy direct influence matrix.

**Table 5.** Degree of influence-relation and the corresponding fuzzy number.

| Degree of Influence-Relation | Intuitionistic Fuzzy Values |
|---|---|
| Very Strong (VS) | (0.95, 0.05, 0.00) |
| Strong (S) | (0.80, 0.15, 0.05) |
| Moderate (M) | (0.65, 0.25, 0.10) |
| Weak (W) | (0.50, 0.35, 0.15) |
| Very Weak (VW) | (0.40, 0.40, 0.20) |

After weighting and defuzzification using Equations (6) and (7), the direct influence matrix *B* was obtained.

$$
B = \begin{bmatrix}
b_{11} & \cdots & b_{1p} & \cdots & b_{1m} \\
\vdots & \ddots & & & \vdots \\
b_{p1} & & b_{pq} & & b_{pm} \\
\vdots & & & \ddots & \vdots \\
b_{m1} & \cdots & b_{mq} & \cdots & b_{mm}
\end{bmatrix}
\tag{8}
$$

where $b_{pq}$ ($p$, $q$ = 1, 2 ... $m$) represents the influence of factor RFp on factor RFq. When $p = q$, $b_{pq} = 0$.

(2)  Normalization of the direct-relation matrix *M*:

$$
M = \mu \times B
\tag{9}
$$

$$
\mu = 1 / \max_{0 \leq p \leq m} \left( \sum_{q=1}^{m} b_{pq} \right)
\tag{10}
$$

(3)  Calculation of the total relation matrix *T*:

$$
T = \left[ t_{pq} \right]_{m \times m} = M(I - M)^{-1}
\tag{11}
$$

where *I* is the identity matrix.

(4)  Influence–importance determination of RFs

$E_p$ is the summation of elements in the pth row of matrix *T*, which indicates the influence strength of $RF_P$ on other $RFs$, both directly and indirectly. $F_q$ is the summation of elements in the *p*th columns of matrix *T*, which indicates the influence strength that the other $RFs$ have on $RF_p$, both directly and indirectly.

$$
E_p = \sum_{q=1}^{m} t_{pq}
\tag{12}
$$

$$
F_q = \sum_{p=1}^{m} t_{pq}
\tag{13}
$$

when $p = q$, $PR_p$ is the summation of $E_p$ and $F_q$, which comprehensively reflects the relationship between a factor and other factors, and represents the degree of effect of the factor in the system. Because the degree $RE_p$ is the difference between $E_P$ of the impact and $F_q$ of the impact, if $RE_p$ is greater than 0, it indicates that the factor has a great influence on other factors, and it is then called a cause factor. If $RE_p$ is less than 0, it indicates that the factor is greatly influenced by other factors, and it is then called a result factor.

$$
PR_p = E_p + F_q
\tag{14}
$$

$$
RE_p = E_p - F_q
\tag{15}
$$

Based on centrality $PR_p$ and reason $REp$, the relative importance of $RF_p$ can be determined.

$$
FIw_p = \sqrt{PR_p^2 + RE_p^2}
\tag{16}
$$

$$
FIW_p = \frac{FIw_p}{\sum_{p=1}^{m} FIw_p}
\tag{17}
$$

where $FIW_p$ represents the relative importance of $RF_p$ compared to other risk factors.

### 3.5. Construction of a Two-Stage HoQ Model

A two-stage HoQ model is constructed based on the QFD method, connecting CRs with RFs and RFs with RESs, respectively. The framework of HoQ is shown in Figure 4.

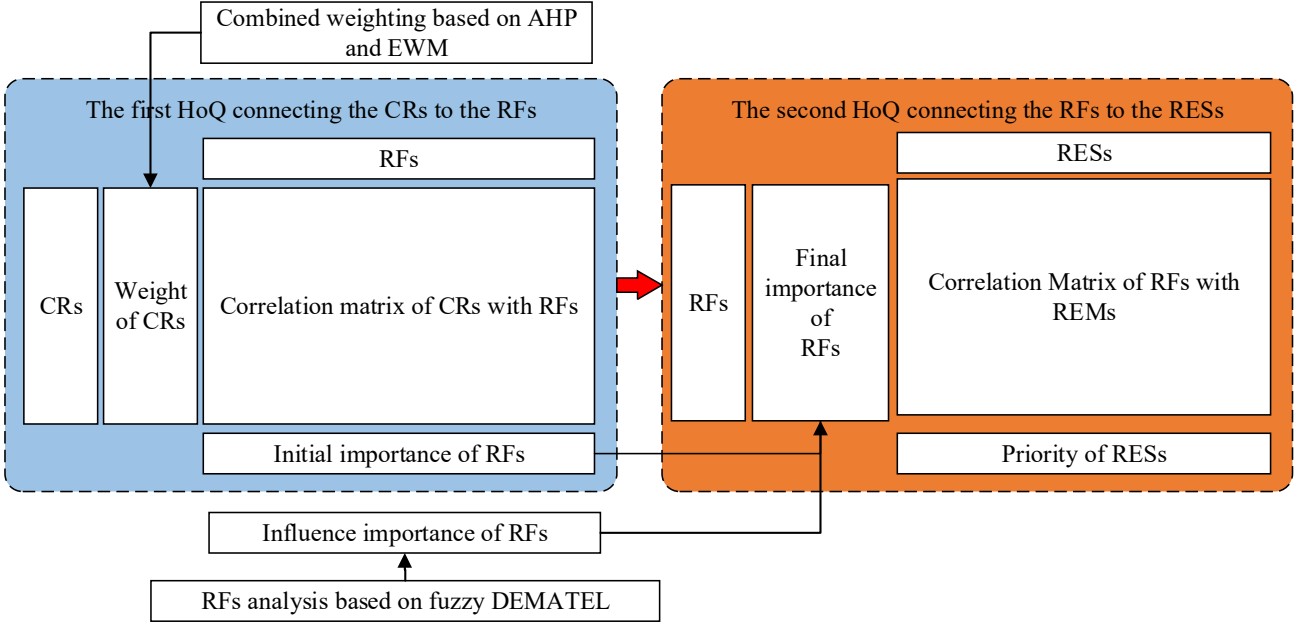

**Figure 4.** Structure of the two-stage HoQ model.

Through the first HoQ, the basic severity of RFs can be obtained from Equation (18).

$$RFI_p = \sum_{j=1}^{u} D_j C_{jp} \tag{18}$$

where $C_{jp}$ is the correlation degree between the $j$th CR and the $p$th RF in the association matrix Cum.

Combined with the analysis results of fuzzy DEMATEL, the comprehensive severity of $RF_p$ is determined by Equation (19).

$$RFW_p = \alpha FIW_p + \beta RFI_p \tag{19}$$

where $\alpha$ and $\beta$ are the basic severity degree and relative severity degree, respectively, which are determined by DMs, according to specific conditions.

After the comprehensive importance of RFs is obtained, the second HoQ model is constructed according to the correlation between RFs and RESs, and the priority order of RESs is obtained from Equation (20).

$$RESW_l = \sum_{p=1}^{m} RFW_P E_{pl} \tag{20}$$

where $Ep_l$ is the correlation degree between the $p$th RF and $l$th RES in the correlation matrix Emo.

## 4. Results and Discussion

### 4.1. Case Study Results

The method proposed in this paper is applied to analyze the severity of RFs in the BMSC involving Guinea–China, and to prioritize the proposed RESs. First of all, the CRs, RFs and RESs are determined by analyzing the BMSC's actual business requirements and

the relevant data, through research. Then, the relevant data are collected through expert interviews and email questionnaires, and converted into a quantitative value analysis through the model. The five DMs consulted in this paper are three practitioners of BMSC and two scholars who have important influence in the field of maritime supply chain management.

4.1.1. Construction the First HoQ for CRs and RFs

(1) Calculating the weight of CRs. The Delphi method is used for hierarchical analysis, the collected opinions are processed using a geometric average method, and the judgment matrix of pairwise comparison is constructed. The maximum characteristic root λmax is calculated as 5.2971, using MATLAB software (2018). The consistency ratio is 0.0663, less than 0.1, thereby satisfying the requirements of the consistency check. The entropy method is used to calculate the objective weight, and the comprehensive weight is calculated according to Formula (5). The weight calculation results of CRs are shown in Table 6.

**Table 6.** The weight calculation results of CRs.

|  | $W_j$ | $H_j$ | $D_j$ |
|---|---|---|---|
| CR1 | 0.3283 | 0.2285 | 0.2790 |
| CR2 | 0.1019 | 0.1520 | 0.1267 |
| CR3 | 0.1766 | 0.2466 | 0.2126 |
| CR4 | 0.3476 | 0.2691 | 0.3115 |
| CR5 | 0.0456 | 0.1038 | 0.0702 |

(2) Calculating the basic importance of RFs. On the basis of the evaluation of the correlation between CRs and RFs by DMs, the fuzzy linguistics term is converted into an intuitive fuzzy set according to Table 3, and the final correlation matrix is constructed after the deblurring based on the weighted average of Equations (6) and (7). Based on the weight calculation results of CRs, the basic severity of RFs is calculated according to Equation (18). The results of the first HoQ are shown in Table 7.

**Table 7.** The first HoQ model connecting the CRs to the RFs.

| CRs | $D_j$ | RF1 | RF2 | RF3 | RF4 | RF5 | RF6 | RF7 | RF8 | RF9 | RF10 | RF11 | RF12 | RF13 | RF14 | RF15 |
|---|---|---|---|---|---|---|---|---|---|---|---|---|---|---|---|---|
|  |  | | | | | | | Normalized Correlation Matrix between *RFs* and *CRs* | | | | | | | | |
| CR1 | 0.279 | 1.558 | 1.383 | 2.321 | 1.714 | 1.714 | 1.714 | 1.452 | 1.271 | 1.383 | 1.383 | 1.271 | 1.452 | 1.383 | 1.860 | 1.207 |
| CR2 | 0.126 | 1.551 | 1.151 | 1.551 | 1.551 | 1.714 | 1.714 | 1.151 | 0.972 | 1.271 | 2.141 | 1.271 | 2.034 | 2.451 | 2.136 | 1.452 |
| CR3 | 0.213 | 0.841 | 1.860 | 0.898 | 0.898 | 1.017 | 2.451 | 1.093 | 0.841 | 2.034 | 1.271 | 2.321 | 0.972 | 0.841 | 1.391 | 1.860 |
| CR4 | 0.312 | 1.860 | 1.383 | 1.860 | 1.714 | 1.860 | 1.551 | 2.321 | 1.860 | 1.383 | 1.271 | 1.271 | 1.714 | 1.271 | 1.860 | 1.714 |
| CR5 | 0.070 | 1.383 | 2.034 | 1.551 | 1.383 | 1.551 | 2.451 | 1.017 | 1.383 | 1.383 | 1.452 | 2.321 | 1.278 | 1.551 | 1.452 | 1.955 |
|  | RFIp | 1.486 | 1.501 | 1.723 | 1.497 | 1.600 | 1.872 | 1.578 | 1.333 | 1.507 | 1.425 | 1.568 | 1.493 | 1.380 | 1.767 | 1.587 |

(3) Determine the correlation between RFs and their comprehensive importance. According to the evaluation criteria of the interaction relationship in Table 8, the autocorrelation matrix of RFs is constructed, and the relative importance and comprehensive importance of RFs are calculated from Equations (9)–(17). The calculation results are directly presented in Table 7. Therefore, the comprehensive severity of RFs in the BMSC is ranked as follows: RF14 > RF6 > RF13 > RF15 > RF10 > RF5 > RF12 > RF3 > RF2 > RF7 > RF1 > RF11 > RF4 > RF9 > RF8.

**Table 8.** Severity analysis and ranking of RFs.

| RFs | $E_p$ | $F_p$ | $PR_p$ | $RE_p$ | $FIW_p$ | $RFI_p$ | $RFW_p$ | Severity Ranking |
|------|--------|--------|---------|---------|----------|----------|----------|-------------------|
| RF1 | 10.918 | 9.911 | 20.829 | 1.007 | 0.064 | 0.064 | 0.064 | 11 |
| RF2 | 11.781 | 9.608 | 21.389 | 2.173 | 0.066 | 0.064 | 0.065 | 9 |
| RF3 | 9.569 | 9.397 | 18.966 | 0.172 | 0.058 | 0.074 | 0.066 | 8 |
| RF4 | 10.725 | 9.295 | 20.020 | 1.430 | 0.062 | 0.064 | 0.063 | 13 |
| RF5 | 11.734 | 9.521 | 21.255 | 2.213 | 0.066 | 0.069 | 0.067 | 6 |
| RF6 | 10.963 | 11.929 | 22.892 | −0.966 | 0.070 | 0.080 | 0.075 | 2 |
| RF7 | 10.988 | 8.925 | 19.913 | 2.063 | 0.062 | 0.068 | 0.065 | 10 |
| RF8 | 9.055 | 9.016 | 18.071 | 0.039 | 0.056 | 0.057 | 0.056 | 15 |
| RF9 | 10.832 | 8.974 | 19.806 | 1.858 | 0.061 | 0.065 | 0.063 | 14 |
| RF10 | 10.215 | 13.641 | 23.856 | −3.425 | 0.074 | 0.061 | 0.068 | 5 |
| RF11 | 9.743 | 9.763 | 19.506 | −0.020 | 0.060 | 0.067 | 0.064 | 12 |
| RF12 | 10.250 | 12.170 | 22.420 | −1.920 | 0.069 | 0.064 | 0.067 | 7 |
| RF13 | 10.526 | 15.652 | 26.178 | −5.126 | 0.082 | 0.059 | 0.071 | 3 |
| RF14 | 12.822 | 12.352 | 25.174 | 0.470 | 0.077 | 0.076 | 0.077 | 1 |
| RF15 | 11.731 | 11.697 | 23.428 | 0.034 | 0.072 | 0.068 | 0.070 | 4 |

### 4.1.2. Construction the Second HoQ for RFs and REMs

According to the evaluation results of the association relationship between RFs and RESs by DMs, the fuzzy linguistic term is converted into an intuitive fuzzy set according to Table 9, and the final association matrix of RFs and RESs is constructed after the deblurring of weighted average according to Equations (6) and (7). Combined with the comprehensive importance $RFW_p$ of $RFs$ in Table 8, the importance of RESs is obtained according to Equation (20). The results are shown in Table 9, and the priority of RESs is RES6 > RES4 > RES5 > RES2 > RES7 > RES3 > RES1.

**Table 9.** The second HoQ model connecting the RFs to the REMs.

| RFs | *RESs* | RES1 | RES2 | RES3 | RES4 | RES5 | RES6 | RES7 |
|------|---------|------|------|------|------|------|------|------|
|  | $RFW_p$ | Normalized Correlation Matrix between RFs and RESs | | | | | | |
| RF1 | 0.064 | 1.09 | 1.86 | 0.84 | 1.33 | 0.84 | 0.91 | 1.09 |
| RF2 | 0.065 | 1.15 | 0.90 | 0.90 | 0.84 | 0.84 | 1.02 | 1.71 |
| RF3 | 0.066 | 2.33 | 1.71 | 2.45 | 2.45 | 1.39 | 2.45 | 1.20 |
| RF4 | 0.063 | 1.45 | 2.03 | 2.45 | 1.15 | 1.27 | 2.45 | 0.84 |
| RF5 | 0.067 | 1.45 | 2.03 | 2.45 | 2.45 | 1.27 | 2.45 | 1.20 |
| RF6 | 0.075 | 0.84 | 1.20 | 2.45 | 0.90 | 1.09 | 2.21 | 2.03 |
| RF7 | 0.065 | 2.45 | 2.03 | 0.84 | 1.39 | 1.63 | 2.15 | 1.20 |
| RF8 | 0.056 | 0.97 | 1.38 | 0.84 | 1.27 | 0.84 | 1.27 | 1.20 |
| RF9 | 0.063 | 0.90 | 0.84 | 0.84 | 0.90 | 0.84 | 0.84 | 1.38 |
| RF10 | 0.068 | 1.15 | 1.39 | 1.09 | 2.15 | 2.15 | 1.83 | 1.20 |
| RF11 | 0.064 | 0.97 | 0.84 | 0.84 | 0.84 | 0.84 | 0.84 | 1.38 |
| RF12 | 0.067 | 0.90 | 1.71 | 0.90 | 2.45 | 2.33 | 2.33 | 1.27 |
| RF13 | 0.071 | 1.02 | 1.55 | 1.09 | 2.15 | 2.33 | 2.33 | 1.27 |
| RF14 | 0.077 | 1.20 | 0.90 | 1.39 | 2.15 | 2.45 | 2.45 | 2.15 |
| RF15 | 0.070 | 1.45 | 2.03 | 1.02 | 1.71 | 1.86 | 1.86 | 2.15 |
| | RESWo | 0.121 | 0.140 | 0.130 | 0.153 | 0.140 | 0.180 | 0.136 |
| | Priority | 7 | 4 | 6 | 2 | 3 | 1 | 5 |

### 4.2. Discussion

A hybrid framework is developed to connect CRs with RFs and RESs in the BMSC. The results provide insight into the critical CRs for the operational objectives of the BMSC, how RFs impact CRs, and what RESs should be prioritized to alleviate risks. The calculation results show that stability and reliability are the most significant CRs, which is in line with the BMSC's current business situation and the management objectives of the international energy supply chain. In the face of increasing fluctuations in the structure of global energy

demand and increasing variables in the risk of maritime energy supply, it is inevitable that the stability of the BMSC will be promoted. A stable BMSC can effectively avoid the vulnerability that triggers the multi-stage characteristics of the BMSC. On the other hand, ensuring reliable operation between all links of the supply chain is the second largest customer requirement of the BMSC. Smart supply chains have shifted from economy to reliability in recent years, especially in the field of port and shipping. Finally, due to the fluidity of bauxite and the risk of marine operations, improving logistics security is also an important factor in maintaining the activities of the BMSC.

In the severity analysis of RFs, the errors of severity values and ranking are not significantly different in the above three types. The comprehensive severity value considers both objectivity and relevance, and is located between basic severity and relative severity, narrowing the gap between the two. Finding a compromise, it can fully reflect the severity of RFs, and can also reflect the applicability and accuracy of the model results. The severity and ranking of RFs are shown in Figure 5. It is worth noting that in the assessment of basic severity, the lack of bauxite mine modernization in terms of processed technology (RF3) has high severity, but low relative severity, mainly because it is objective. Although RF3 itself has a great impact on supply chain customer requirements, it has little correlation with other RFs. According to the ranking results of comprehensive severity, we can obtain that information sharing asymmetry (RF14), poor ship stability or obsolete equipment performance (RF6), lack of coordination between shipping and ports (RF13), and poor emergency response practices (RF15) are the most serious RFs in the BMSC, which have the greatest impact on CRs. Among them, information sharing asymmetry affects the efficiency of cooperation among participants. The moisture content of bauxite is high, and it is not strictly tested before shipment, which will lead to the formation of a free surface effect in the process of bumpy transport on the sea, thus reducing the stability of the ship. Furthermore, the ship's operating facilities and aging equipment, a lack of maintenance, and the ship sailing without prior inspection may cause cargo loss or hull capsizing. In addition, it is worth pointing out that port handling and shipping are the pivotal activities of the BMSC; risks such as port congestion, cargo delivery delays and resource allocation disorder caused by improper port and shipping cooperation could seriously affect supply chain operations.

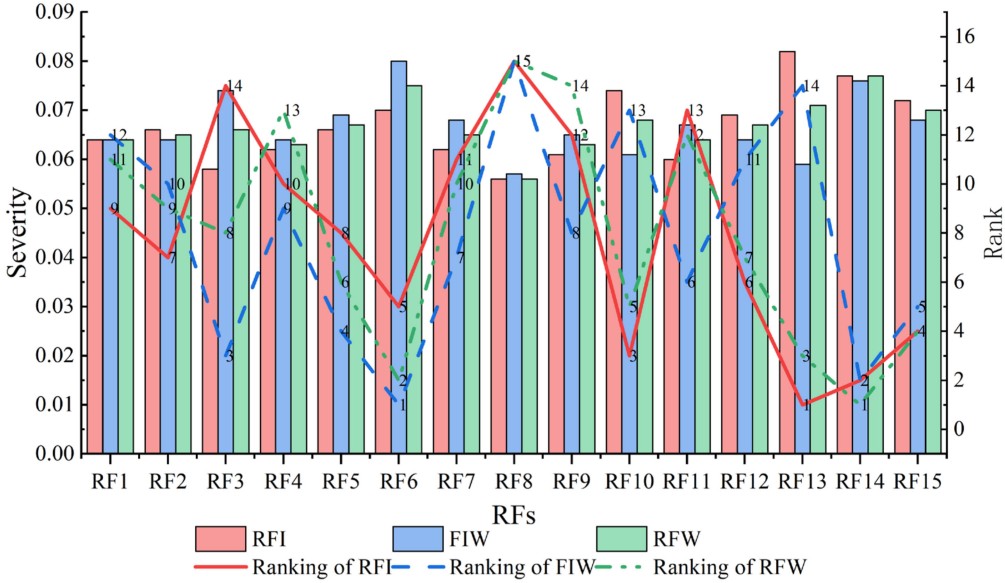

**Figure 5.** Comparison of severity and rank of RFs.

Constructing strategic alliances is the most optimal RESs to mitigate critical risks and improve customer satisfaction. The priority of RES6 is reflected in its significant contribution to all RFs mitigation and CRs implementation. Ports, shipping companies and various

logistics service providers should integrate port and shipping services, integrate upstream and downstream resources of the supply chain, and promote sustainable development of the BMSC. This paper proposes that a cooperative alliance of the BMSC can be constructed in the following ways: (1) All parties of the BMSC industry, including buyers, mines, ports and shipping enterprises, must fully combine their resources and technical advantages to build a comprehensive and cooperative project. For instance, Rotterdam Port, together with its subordinate shipping companies, has established a petrochemical strategic alliance system with various petrochemical giants. (2) Ports require shipping companies to jointly develop the terminal, and the shipping companies to invest in the infrastructure of the port to enjoy the exclusive services of the port; on the other hand, the shipping companies must also participate in investment in the shares of the port. For example, Mediterranean Lines signed a 30-year concession agreement with ABU Dhabi Ports for the Caliphate Port Terminal, and invested $1.1 billion to expand the terminal platform, dredge and deepen berths, and build a regional container-handling center. The desired operation model of the BMSC should be integrated from both vertical and horizontal dimensions, as shown in Figure 6. From the perspective of supply chain management, the essence of building an alliance within the BMSC is building a service supply chain by virtue of enterprise, resources, and technical advantages. All parties must work together to resist uncertain interference events and satisfy customer needs through division of labor.

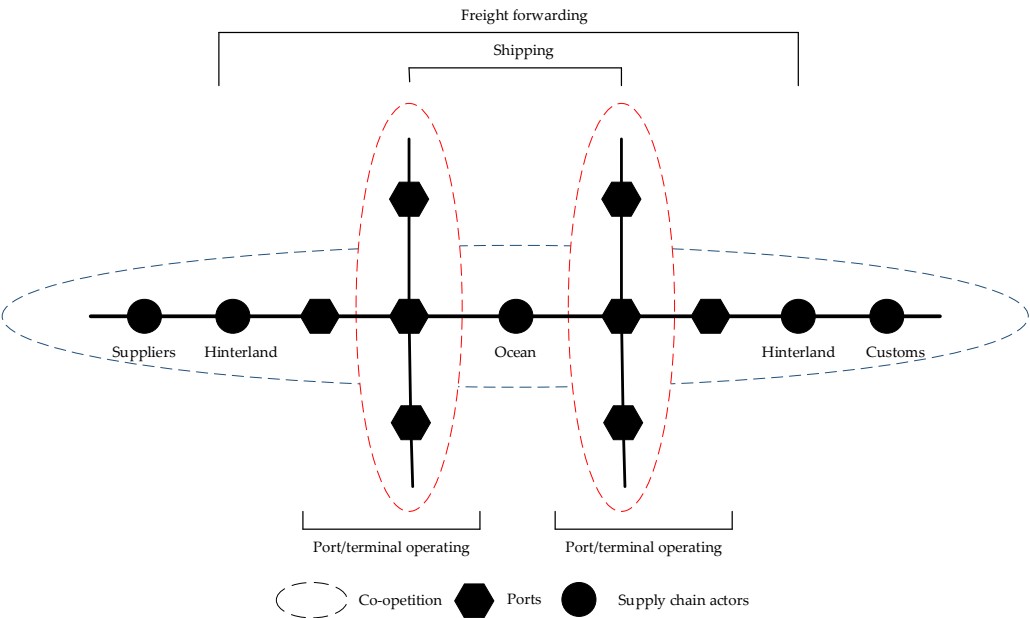

**Figure 6.** The horizontal and vertical integration model of the BMSC under an alliance.

## 5. Conclusions

In this study, a QFD-MCDM with intuitionistic fuzzy decision approach is proposed to integrate the relationships between the CRs, RFs and RESs of the BMSC, which effectively solves the problems of ambiguity and multi-objectivity in the decision process and ensures the effectiveness and accuracy of decision results. Furthermore, the methodology proposed in this paper addresses two successive tasks by constructing a two-stage HoQ; the RFs are assessed after analyzing the CRs, and subsequently, the RESs are decided upon to mitigate the impact of risks and to fulfill operational objectives. From the perspective of the supply chain, the paper draws a conclusion on the severity of RFs and the priority of RESs to broaden the perspectives of risk resilience management in the BMSC. Through analysis, supply chain managers can identify the most important customer requirements, understand the internal connections between risk factors, and perceive the intensity of their interference in the performance of different supply chains. Managers can effectively deploy strategic resources in the case of limited resources and can clearly understand how RESs

can alleviate risk, so as to enhance the resilience of the BMSC. According to our empirical investigation of the Guinea–China bauxite import supply chain, the main conclusions are as follows:

- Stability is the most critical goal of the BMSC operation;
- The top three RFs affecting the CRs are 'information sharing asymmetry', 'poor ship stability or obsolete equipment performance', and 'lack of coordination between shipping and ports';
- The RES that can minimize risks and improve performance is 'constructing a strategic alliance'; horizontal and vertical integration of the BMSC can be adopted as the main mode of alliance.

The present study also has some limitations. For instance, this study selected the optimal RESs only from the perspectives of risk and operation. Future studies will further consider the enforcement cost and performance after implementation of each RES, determine the resilient index in multiple dimensions, and determine the optimal construction scheme of the BMSC based on moderate resilience. Additionally, according to the different scenarios of the BMSC, carrying out resilience measurements by constructing an index and sustainable performance criteria will be an interesting research direction for future works.

**Author Contributions:** Conceptualization, H.W.; methodology, J.S. and H.W.; data curation, J.S. and Z.C.; validation, H.W., J.S. and Z.C.; investigation, J.S. and Z.C.; writing—original draft preparation, J.S.; writing—review and editing, H.W.; visualization, J.S. and Z.C.; supervision, H.W.; project administration, H.W. All authors have read and agreed to the published version of the manuscript.

**Funding:** This research was funded by the National Natural Science Foundation of China, grant number 51909202.

**Institutional Review Board Statement:** Not applicable.

**Informed Consent Statement:** Not applicable.

**Data Availability Statement:** Not applicable.

**Acknowledgments:** The authors would like to thank those who provided help in improving the work. The authors also thank the editors and the anonymous reviewers for their constructive suggestions for improving the paper.

**Conflicts of Interest:** The authors declare no conflict of interest.

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
