# Peer review of "Alleviating the Bauxite Maritime Supply Chain Risks through Resilient Strategies: QFD-MCDM with Intuitionistic Fuzzy Decision Approach"

_sustainability, doi:10.3390/su15108244_

Round 1
Reviewer 1 Report
1. I consider that the topic is actual and scientifically interesting. The manuscript is clearly structured and organized, it is easy to follow and the terminology is appropriate to the subject. Tables and figures are used efficiently and support the text, also reference citations are complete and accurate.
2. The content of the paper is succinctly described and contextualized in relation to the presented theoretical background.
3. I would recommend to emphasize in the abstract more strongly the relevance, originality and quality of the research, persuasively suggesting to the potential reader the items of interest that the work proposes.
4. I recommend the authors to present much more clearly and articulately the context in which the problem is formulated in the introductory part of the paper. In my opinion this is a good scientific paper, but looks like maritime transportation management. The authors must explain and detailed why this subject is relative to Sustainability and how does this study contribute to Sustainability?
5. In my opinion in Section 3 Methodology, the research method which was used is presented clearly and in detail, providing the necessary elements for the reproduction of research by any other research group that uses it exactly (the repetitive and reproducible nature of science).
6. The results and data obtained must be robust enough to draw conclusions, we recommend that you review the Section Results and Discussion.
7. The concluding elements of the paper are represented by strong statements based on scientific arguments that are presented clearly and concisely. However, I believe that the authors should reflect the extent to which the results answered the questions mentioned in the introductory part. In my opinion, the solid argumentation of the conclusions of the paper will open new research directions and lead to the deepening of the issues studied by potential readers.
8. We have found that bibliographic references are described accurately, honestly and deontologically by the authors.
9. I recommend the authors to develop/ to detail the future research opportunities considered feasible and scientifically fertile in the field.
10. Minor corrections: Renumber the steps of subsection 4.1.1 (line 514); bpq or bpq (lines 440, 441); Ep or Ep, Fp or Fp / (lines 449, 450, 451) etc etc.
Author Response
We sincerely thanks for your valuable feedback that we used to improve the quality of our manuscript. We look forward to hearing from you regarding our submission and be glad to respond to any further questions and comments that you may have.Please see the attachment for detailed modification instructions.

Reviewer 2 Report
The paper titled as "Alleviating the Bauxite Maritime Supply Chain Risks through Resilient Strategies: QFD-MCDM with Intuitionistic Fuzzy Decision Approach" presents an integrated methodology to achieve efficient design of BMSC resilient strategies (RESs).
There are research question in Introduction section - what is the basic element of the good research.
Literature review involves related research background, including literature sources which are up to date.
It will be good to provide, e.g., theoretical example using which all the formulas will be explained as there are only formulas and results - any example/explanation how to apply it.
The discussion is comprehensive and cover all aspects related to the discussion.
Methodology is well written and graphically explained.
Please add into the conclusion section why is your research better than existing approaches.
Author Response

(The authors gave the same response as above.)

Reviewer 3 Report
The overall quality of the paper is good and worth recommending, but the quality of the English language needs further modification.
The language of English should be concise and conform to English grammar and the logical rules of writing.It is recommended to read the entire text thoroughly and make careful modifications.
Author Response

(The authors gave the same response as above.)
